# Spore Oil-Functionalized Selenium Nanoparticles Protect Pancreatic Beta Cells from Palmitic Acid-Induced Apoptosis via Inhibition of Oxidative Stress-Mediated Apoptotic Pathways

**DOI:** 10.3390/antiox12040840

**Published:** 2023-03-30

**Authors:** Sajin Zhou, Hongyan Zhu, Piaopiao Xiong, Lei Shi, Weibin Bai, Xiaoling Li

**Affiliations:** 1Institute of Food Safety and Nutrition, Jinan University, Guangzhou 510632, China; 2Guangdong Engineering Technology Center of Molecular Rapid Detection for Food Safety, Jinan University, Guangzhou 510632, China

**Keywords:** palmitic acid, spore oil nano-selenium complex, pancreatic β-cells, oxidative stress, apoptosis

## Abstract

Oxidative stress damage of pancreatic β-cells is a key link in the pathogenesis of type 2 diabetes mellitus. A long-term increase of free fatty acids induces the increase of reactive oxygen species (ROS) in β-cells, leading to apoptosis and dysfunction of β-cells. *Ganoderma lucidum* spore oil (GLSO) is a functional food complex with strong antioxidant activity, but its solubility and stability are poor. In the present study, GLSO-functionalized selenium nanoparticles (GLSO@SeNPs) with high stability and uniform particle size were synthesized by a high-pressure homogeneous emulsification method. The aim of this study was to investigate the protective effects of GLSO@SeNPs on INS-1E rat insulinoma β-cells against palmitic-acid (PA)-induced cell death, as well as the underlying mechanisms. Our results showed that GLSO@SeNPs had good stability and biocompatibility, and they significantly inhibited the PA-induced apoptosis of INS-1E pancreatic cells by regulating the activity of related antioxidant enzymes, including thioredoxin reductase (TrxR), superoxide dismutase (SOD), catalase (CAT), and glutathione peroxidase (GSH-Px). Western blot analysis showed that GLSO@SeNPs reversed the PA-induced changes in MAPK pathway protein expression levels. Thus, the present findings provided a new theoretical basis for utilizing GLSO@SeNPs as a treatment for type 2 diabetes.

## 1. Introduction

A global diabetes map was published on the official website of the International Diabetes Federation (IDF), which showed that the number of adult patients with diabetes in the world had reached 537 million; in addition, it has been predicted that the number of adult patients with diabetes in the world will exceed 643 million by 2030 and reach 783 million by 2045 [1], indicating that diabetes around the world is serious.

Studies have shown that obesity is more likely to induce type 2 diabetes [2,3]. Because obese individuals are in a high-fat state, the increase of free fatty acid content in this state causes insulin resistance and reduces the number of pancreatic β-cells by apoptosis, leading to abnormal insulin secretion and decreased expression of insulin-receptor-related proteins. This phenomenon is called lipotoxicity [4,5]. Lipotoxicity is related to inflammation and oxidative stress; it not only acts through some metabolic pathways but also affects some protein receptor pathways in cells [6,7]. However, the ability of pancreatic β-cells to resist oxidative stress is relatively weak, and oxidative stress induced by being in a high-fat state inevitably causes damage to the function and number of β-cells. Palmitic acid (PA), a saturated fatty acid [8,9] widely present in palm, vegetable, and catalpa oils, is the major component obtained from the daily diet that accumulates in the body, resulting in lipotoxicity of pancreatic β-cells.

*Ganoderma lucidum* spore oil (GLSO) is mainly derived from *Ganoderma lucidum*, which has anti-tumor, immunomodulatory, and anticerebral ischemia/reperfusion injury effects [10,11,12] as well as the ability to scavenge oxidizing free radicals in the body. The main biologically active components of GLSO are unsaturated fatty acids, triterpenoids, polysaccharides, and steroids [13,14]. In the process of the bioutilization of GLSO, these functional components are preferentially oxidized, which prevents the organism from being attacked by oxidizing free radicals, thereby playing an antioxidant role. Diabetes is a metabolic disease characterized by hyperglycemia. One study showed that [15] *Ganoderma lucidum* contains a variety of bioactive compounds, such as polysaccharides, proteins, and polyphenols, which can control blood sugar and be used in anti-cancer treatment. *Ganoderma lucidum* has the nutritional and medicinal potential to combat gestational diabetes (GDM) by altering sugar metabolism and reducing lipid peroxidation. It provides valuable information for understanding the pathogenesis and treatment direction of diabetes.

However, the application of GLSO is limited due to its instability and low solubility. Recently, the continuous development of nanotechnology has provided a new direction for the wide application of GLSO. Nanodrug delivery systems constructed by natural carrier modifications have been widely used in the delivery process of traditional drugs into the body, making traditional drugs more efficient with accurate delivery of the therapeutic target, thereby improving the bioavailability of drugs [16]. Some researchers have explored the protective effect of GLSO nanoparticles (NPs) on myocardial injury caused by thoracic radiotherapy [17]. Selenium is an essential trace element in the human body, and selenium exerts its effects in the human body mainly through the formation of selenoproteins, such as selenocysteine (Sec) and selenomethionine (Se-Met) [18]. The synthesis of selenoproteins is a prominent reason why selenium exerts good free-radical-scavenging abilities in vivo. Some researchers have designed and developed selenium-layered NPs (SeNPs) and combined them with mulberry leaf and pueraria root extracts for oral administration, promoting insulin secretion to inhibit the increase of blood glucose levels and improve pancreatic function [19]. More interestingly, SeNPs can also be used as a carrier to synthesize unique oxidation inhibitors, which can prevent free-radical-induced pancreatic damage by inhibiting the production of reactive oxygen species (ROS) in cells [20]. GLSO and SeNPs have strong antioxidant activity, indicating that they have potential application prospects in antagonizing the lipid toxicity of pancreatic β-cells.

In the past few years, some studies have explored the bioactivity of GLSO in inhibiting the apoptosis of tumor cells [21], achieving good results. However, few studies have used nanotechnology to load and modify GLSO to improve the bioavailability and stability of materials. Therefore, to obtain the highest therapeutic potential of GLSO in food and medicine, we introduced the common GLSO into the nanosystem by selecting biodegradable polymeric materials and using a high-pressure homogenization emulsification method in the present study. The aim of the present study was to improve the solubility and bioavailability of GLSO by utilizing SeNPs to realize dual antioxidant activity in antagonizing the lipid toxicity of pancreatic β-cells. After morphological characterization and biocompatibility evaluation of the synthesized GLSO@SeNPs, we established a PA-induced INS-1E β-cell lipid toxicity model and conducted cell viability, cell cycle, mitochondrial membrane potential, ROS, antioxidant enzyme activity, intracellular selenium metabolism, western blot, and other assays.

## 2. Materials and Methods

### 2.1. Materials

RPMI-1640 medium, fetal bovine serum (FBS), and an antiseptic solution (penicillin and streptomycin) were obtained from Gibco (New York, NY, USA). Propidium iodide (PI) and PA(76119-5g) were purchased from Sigma-Aldrich (St. Louis, MO, USA). Hoechst 33342 (BL803A) was bought from a biosharp supplier. β-mercaptoethanol (BME) was purchased from Macklin (Shanghai, China). CCK-8(C0039), JC-1 mitochondrial membrane potential (C2006), Mito-Tracker (Green) (C1048), and glutathione peroxidase (GPx) assay kits (S0058) were obtained from Beyotime Corporation (Shanghai, China). Thioredoxin reductase (TrxR) (BC1150) and catalase (CAT) assay kits (BC4785) were purchased from Solarbio Technology Company (Beijing, China). The antibodies used for western blot analysis, such as C-PARP (94885), PARP (9532T), p-AKT (#4060S), AKT (#4091S), P-p44/42(#4370S), t-ERK/p44/4-2 (#9102), p-38 (#8690T), p-p38 (#9211S), JNK (#9252), p-JNK (9251S), p-53 (#252-4S), p-p53 (#9286S), C-Caspase-3 (#9661), Caspase-8 (#4790S), Caspase-10 (#9752), Bim (#2933), and β-actin (3700T) were acquired from Cell Signaling Technology (Danvers, MA, USA). Bad (1541-1) was from Epitomics, Bax (A19684) was from Abclon-al, and Bcl-2 (T40056S) was from Abmart (Shanghai, China). GLSO was gifted from another laboratory. Oxidative stress indicator assay kits for superoxide dismutase (SOD) (A001-3-2), malondialdehyde (MDA) (A003-1-2), and reduced glutathione (GSH) (A006-2-1) were purchased from Jiancheng Biological Company (Nanjing, China).

### 2.2. Synthesis and Characterization of GLSO Nanoemulsions (GLSO@NEs), SeNPs, and GLSO@SeNPs

GLSO@NEs were synthesized based on a previously published method [17] with slight modification. In brief, Tween 80 (3 mL) and ethanol (1 mL) were mixed for 5 min, and 5 mL of GLSO was then added, followed by gentle stirring for 5 min to obtain a GLSO primary emulsion after adding H_2_O to the above mixture to a final volume of 50 mL. Subsequently, the high-pressure homogenizer was employed to prepare a GLSO NE using the following homogeneity conditions: homogenization pressure of 200 bar for 2 min, 1200 bar for 10 min, and a sampling frequency of 40 Hz.

SeNPs were synthesized via an ascorbic acid (Vc) reduction method according to previously published methods [22]. Firstly, a 20 mM sodium selenite (Na_2_SeO_3_) solution, an 80 mM Vc solution, and a 2 mg/mL lentinan (LET) solution were prepared using ultrapure water. Next, 4 mL of Na_2_SeO_3_ was mixed with 12 mL of LET solution for 5 min, and 4 mL of Vc solution was then slowly added in a dropwise manner for reduction. After 12 h, the above solutions were dialyzed through dialysis bags (6000–8000 D) for 2 days, and the SeNPs were then collected.

Finally, a certain mass of Poloxam 407 powder was added to a 50 mL beaker, and certain concentrations of SeNPs and GLSO@NEs were added. Distilled water was then added to reach a final volume of 30 mL. The mixture was then mixed at 4 °C to produce a final composite gel with a mass fraction of 25%, a concentration of 2 mM SeNPs, and a 5% concentration of GLSO@NEs, which resulted in the GLSO@SeNP hydrogel composite system. These nanomaterials were characterized using transmission electron microscopy (TEM; 120 kV), a nanoparticle size potentiometer (Malvern, Nano-ZS, UK), and Raman spectroscopy (HORIBA, LabRAM HR Evolution). During TEM measurement, 1–2 drops of samples were dropped into the copper mesh and dried completely. Finally, the morphology of the samples was observed under a transmission electron microscope (TEM) (Hitachi H-7650, JEM Flash 1400, 120 Kv, Tokyo, Japan). In the Raman experiment, the three nanomaterials were dried in a dryer at room temperature for 12 h, transferred to a clean glass sheet, and then the samples were measured using a laser Raman microscope under 488 nm laser excitation.

### 2.3. Stability Assessment of GLSO@NEs, SeNPs, and GLSO@SeNPs

The stability of GLSO@NEs, SeNPs, and GLSO@SeNPs was assessed by monitoring their particle size and potential changes in H_2_O, phosphate-buffered saline (PBS), and cell culture medium (without phenolic red and FBS). Certain concentrations of the nanosystems were added to the above solution and placed at 4 °C. Their particle size and potential information were measured at different time points (0, 12, 24, 48, 72, and 168 h).

### 2.4. Hemocompatibility of GLSO@NEs, SeNPs, and GLSO@SeNPs

Healthy human-derived erythrocytes were used to demonstrate the good hemocompatibility of GLSO@NEs, SeNPs, and GLSO@SeNPs. First, erythrocytes were diluted with PBS, aliquoted into test tubes, and coincubated with different nanosystems at 37 °C for a certain time (2, 4, and 8 h). Moreover, a negative (PBS) and a positive (Triton X-100) control group were set up. The morphological characteristics of erythrocytes in each group were recorded with an electron microscope at each time, followed by centrifugation to collect the supernatant for spectrophotometric detection (540 nm) to assess the hemolysis rate of nanodrugs. The hemolysis rate was calculated by the following formula: hemolysis rate (%) = (λ_Nanosystem_ − λ_NG_)/(λ_PG_ − λ_NG_) × 100%, where λ_Nanosystem_, λ_NG_, and λ_PG_ are the absorbance of nanoparticles, negative control group, and positive control group at 540 nm, respectively.

### 2.5. Cell Culture and Cell Viability Assay

The INS-1E rat insulinoma cell line was obtained from another laboratory (Hong Kong). INS-1E cells were cultured with RPMI-1640 medium supplemented with L-glutamine (2 mM), sodium pyruvate (1 mM), HEPES (10 mM), 10% FBS, mercaptoethanol (50 μM), penicillin (100 units/mL), and streptomycin (100 μg/mL) at 37 °C in a humidified incubator with 5% CO_2_. Cell viability was measured using a CCK-8 reagent as described previously [23]. Briefly, INS-1E cells (3 × 10^3^ cells/well) were seeded in 96-well plates and cultured for 24 h. Cells were then treated with the IC_50_ concentration of PA and cytotoxic concentrations of nanosystems. To determine the protective effects of GLSO@NEs, SeNPs, and GLSO@SeNPs, adherent cells were pretreated with different drug concentrations for 6 h followed by treatment with PA and each nanosystem (GLSO@Nes + PA, SeNPs + PA, and GLSO@SeNPs + PA). After 24 h, the cell culture medium was removed, and CCK-8 reagent was added, followed by incubation for 2–4 h. The absorbance was then read using a fluorescence microplate reader (TECAN Infinite F50, Shanghai, China). Images of the cells after drug treatment were acquired using an electron microscope.

### 2.6. Analysis of Cell Cycle Distribution

The cell cycle distribution of INS-1E cells treated with each nanosystem was analyzed using flow cytometry (Beckman Coulter, Miami, FL, USA) as previously reported [24]. First, cells (1 × 10^5^ cells in 5 mL) were seeded into 6 cm culture dishes and allowed to adhere. Cells were then treated with certain concentrations of GLSO, SeNPs, and GLSO@SeNPs for 3 h, followed by treatment with PA for 24 h. Finally, cells and supernatants were collected, centrifuged, and washed two times with PBS. The cell precipitates were suspended in precooled 70% ethanol and incubated overnight at −20 °C to increase cell membrane permeability. On the second day, INS-1E cells were centrifuged, stained with propidium iodide (PI/RNase, 0.05 mg/mL), and analyzed by flow cytometry. At least 10,000 cells were recorded per sample. The analysis of the cell cycle by flow cytometry is based on DNA content (horizontal coordinate): in flow cytometry, the G1 phase is the first peak, the S phase is the second peak (peak span is very large but not high), the G2 phase is the third peak, and the analysis method of DNA content cannot separate the M phase and the G2 phase.

### 2.7. Mitochondrial Membrane Potential (ΔΨm) and Fragmentation Analysis

For the mitochondrial membrane potential experiment, we used the JC-1 probe kit for detection. In brief, after the observation time, the cell samples were digested with trypsin until they fell off the bottom of the petri dish, washed with PBS 1~2 times, and then stained with a JC-1 probe (1 mg/mL) for 20~30 min under light protection. Finally, the distribution ratio of red and green cells was detected using flow cytometry (at least 10,000 cells were detected in each sample).

In the fragmentation experiment, INS-1E cells were treated with drugs as described previously [25]. Cells were then stained with MitoTracker (Green: 0.1 mM) for 20 min and Hoechst 33342 (Blue: 1 μg/mL) for 10 min. The mitochondrial morphology of INS-1E cells was then imaged using a confocal laser scanning microscope (LSM880 with AiryScan, Carl Zeiss, Japan).

### 2.8. Assessment of ROS Generation

INS-1E cells were seeded into black 96-well plates (5 × 10^5^ cells/well) and allowed to adhere. Cells were then pretreated with three nanosystem drugs and PA for 6 h and 12 h. Cells were then incubated with DCFH-DA for 20–30 min, followed by two washes with PBS. The ROS fluorescence values were immediately detected at λex = 488 nm and λem = 525 nm.

### 2.9. Measurement of SOD, MDA, and GSH

SOD, MDA, and GSH assays were performed according to the manufacturer’s instructions (Nanjing Jiancheng Biological Company, Nanjing, China) [26,27]. For the SOD assay, the control, control blank, assay, and assay blank wells were prepared according to the instructions, followed by incubation at 37 °C for 20 min. The absorbance at 450 nm was then read using a microplate reader. For the MDA assay, the blank, standard, assay, and control tubes were prepared according to the instructions followed by centrifugation. The supernatants were added to 96-well plates (200 μL/well), and the absorbance at 532 nm was read using a microplate reader. For the GSH assay, blank, standard, and sample wells were prepared according to the instructions. A volume of 225 μL per well was added to a 96-well plate and allowed to stand for 5 min. The absorbance value at 405 nm was then measured using a microplate reader.

### 2.10. CAT, TrxR, and GPx Activity Assays

The CAT, TrxR, and GPx activity assays were performed according to previously reported methods [28,29,30]. For the CAT activity assay, cells were lysed with a cell lysis buffer, and protein samples were diluted to appropriate concentrations with catalase assay buffer provided with the kit. The absorbance of the blank control group and sample group in a 96-well plate at 240 nm was measured. For the TrxR activity assay, cells were lysed by adding the specified reagent solution for 30 min and then heated in a 37 °C water bath. Blank wells and assay wells were prepared, and the absorbance at 412 nm was measured. The TrxR activity was calculated using the following formula: TrxR (U/mg prot) = 245 × (ΔA_assay tube_ − ΔA_blank tube_)/Cpr. For the GPx assay, cellular proteins were extracted using an ultrasonic ice bath and then mixed with GPx detection buffer and GPx working solution. The blank control group was prepared at the same time. Finally, the absorbance at 340 nm was measured using a microplate reader.

### 2.11. GLSO@SeNP Metabolism in INS-1E Cells

According to a previous research method [31], the control group contained no GLSO@SeNPs or PA, and the experimental groups were treated with GLSO@SeNPs (4 μL/mL) and GLSO@SeNPs (4 μL/mL) + PA (0.3 mM) for 0, 12, 24, 36, and 48 h. After 48 h, the cell supernatant was collected by centrifugation and then detected by high-performance liquid chromatography-inductively coupled plasma mass spectrometry (HPLC-ICP-MS; Hamilton PRP X-100 ion-exchange column). Standards for Se-related metabolites (SeCys2, SeMet, SeIV, MeSeCys, and SeVI) were used for quantitative analysis of GLSO@SeNP metabolites.

The metabolism of GLSO@SeNPs in INS-1E cells was measured by high-performance liquid chromatography-inductively coupled plasma mass spectrometry (HPLC-ICP-MS): First, 1.9213 g citric acid was dissolved in 1L Milli-Q H_2_O (10 mmol/L). Then, the solution was neutralized with 25% ammonium hydroxide (wt) to a pH of 4.50. The eluent was filtered by 0.22 μm microfiltration membrane. The specific equipment detection conditions were as follows: the selenium product was detected for 1500 s by isometric elution at a flow rate of 0.8 mL/min or 0.6 mL/min. SeCys2, SeMet, SeIV, MeSeCys, and SeVI were used as standards at the concentrations of 200, 100, 50, 25, and 0 μg/L to quantify the metabolites of GLSO@SeNPs on Thermo Scientific iCAP™RQ ICP-MS and UltiMate 3000 UHPLC. The collected data were analyzed automatically by Qtegra, and the results were exported to Graphpad 8.0 visualization platform.

### 2.12. Western Blot Analysis

Cells were treated as described above and then lysed using western and IP cell lysis buffer (product no. P0013). Cellular proteins were then analyzed by the conventional western blot analysis method as previously described [32]. Image J software(V1.8.0.112) was used to analyze the protein bands.

### 2.13. Statistical Analysis

GraphPad Prism 7.0 and Origin 8.6 were used for all statistical analyses. In addition, CytExpert(2.4.0.28) and Modfit software(5.0) were used for data processing in the corresponding experiments. All data are presented as the mean ± standard deviation (n ≥ 3). One-way analysis of variance (ANOVA) was used for multiple comparisons among groups. * represents the comparative significance analysis between the control group (unstimulated control) and the modeling group (PA alone treatment), and # represents the comparative significance analysis between the modeling group (PA alone treatment) and the modeling plus protective drug group (GLSO@SeNPs + PA treatment). Among them, * (#) *p* < 0.05 means the lowest significance, ** (**##**) *p* < 0.01 means moderate significance, *** (**###**) *p* < 0.001 means high significance, **** (**####**) *p* < 0.0001 means the highest significance.

## 3. Results

### 3.1. Preparation and Characterization of GLSO@NEs, SeNPs, and GLSO@SeNPs

Three nanomaterials were synthesized according to the above methods, and their particle size and potential were measured by a Malvern particle size analyzer. The morphology of the drug particles was further observed by TEM. As shown in Figure 1B, the newly synthesized GLSO@NEs, SeNPs, and GLSO@SeNPs had particle sizes of 90, 120, and 220 nm, respectively, which were all within the reasonable particle size range [33]. Figure 1C shows that the three-drug particles were all negatively charged, and the charge of GLSO@NEs was close to −20 mV. Greater negative charges represent more repulsive particles, indicating greater difficulty for cluster formation and increased stability. The morphology of the drug particles is shown in Figure 1D. The synthesized SeNPs were circular, well dispersed among the particles, and had a uniform particle size. The GLSO@NEs appeared as a milky liquid with an irregular, circular shape, and they had a uniform particle dispersion with a roughly uniform particle size. In contrast, the GLSO@SeNPs showed a tendency for SeNPs to combine with GLSO@NEs, and the particles were bound together in their original shape by the bonding force of the gel, thus exerting a dual effect. According to the Raman spectrum shown in Figure 1E, due to the large number of carboxyl and hydroxyl functional groups contained in lendinus edoides, the most obvious peak position of lendinus edoides appeared at 1574.75 cm^−1^ (the spectrum usually attributed to the carboxyl group has the C–O antisymmetric stretching vibration peak of 1610~1560 cm^−1^ and C–O symmetric stretching vibration peak of 1420~1394 cm^−1^). A relevant study [34] showed that lentinan-modified SeNPs showed prominent peaks at 1030 cm^−1^ and 1400 cm^−1^. For GLSO@NEs, the characteristic peaks of GLSO appeared at 555.373, 1100.11, 1473.54, and 2880.590 cm^−1^, indicating that the characteristic absorption peaks of GLSO did not shift significantly after the nanominiaturizationof GLSO. GLSO@SeNPs contains a series of prominent peaks shown in the Raman spectra of SeNPs and GLSO@NPs, indicating that GLSO@SeNPs combines the structural features of SeNPs and GLSO@NPs, and thus, GLSO@SeNPs is considered to be their complex. As for the new peak appearing at 3425.45 cm^−1^ in GLSO@SeNPs Raman spectrum, it may be the −OH stretching vibration peak (3400–3200 cm^−1^) contained in poloxam hydrogel.

### 3.2. Stability of GLSO@NEs, SeNPs, and GLSO@SeNPs in Water, PBS, Phenolic Red, and Human Blood

The particle size and potential are the two main indexes used to evaluate the stability of an NP [35]. The changes in particle size and potential of the synthesized SeNPs, GLSO@NEs, and GLSO@SeNPs in some solutions were measured in the present study. Figure 2A,B show that with the change of time, the particle size of SeNPs in water and PBS was stabilized at 129.0 ± 6.2 nm (119.1~138.4 nm). The particle size of GLSO@NEs in water and PBS was stable at approximately 64.5 ± 3.6 nm (61.1~69.8 nm), and the stability range of the particle size of GLSO@SeNPs was slightly higher, ranging from 90.7 ± 13.7 nm (74.4~113.9 nm) in water and PBS. Figure 2C shows that the potentials of the synthesized GLSO@NEs, SeNPs, and GLSO@SeNPs were always in a negative state, which indicated higher stability. From the experimental results of particle size stability, we can see that with the extension of time, the average particle size of nanoparticles increases to a certain extent, which is caused by the partial aggregation of nanoparticles in an aqueous solution and PBS solution. We can see from the experimental results of potential stability that the nanoparticle can still maintain its original negative potential stable state for up to 15 days. The hemolysis ratios of GLSO@NEs, SeNPs, and GLSO@SeNPs were measured to evaluate their blood compatibility. After 2, 4, and 8 h of incubation with red blood cells, the nanodrugs showed a hemolytic concentration of less than 0.1% (Figure 2F). Analysis by light microscopy (Figure 2D) indicated that the red blood cells in the negative control group were round with a smooth, complete surface. Although the surface of red blood cells treated with the three nanodrugs was not as smooth as that of the negative control group, and the number of cells observed in the field of view was slightly less, the red blood cells showed no tendency of lysis with round, smooth cells after drug treatment for 8 h. Figure 2E shows that the red blood cells treated with Triton-X100 were completely ruptured, whereas the red blood cells incubated with the three drugs showed the same morphology as the negative control group.

### 3.3. GLSO@SeNPs Protect against PA-Induced Cytotoxicity

We next detected cell viability via a CCK-8 assay. After treating INS-1E cells for 24 h with various concentrations of PA, the survival rates were 60% and 30% when cells were treated with 0.2 mM and 0.4 mM PA, respectively (Figure 3A). Thus, we selected a PA concentration of 0.3 mM for subsequent experiments. Figure 3B shows that the three nanosystem drugs significantly inhibited PA-induced pancreatic β cytotoxicity at high concentrations in a concentration-dependent manner. The concentration of GLSO@SeNPs at 0.04 μL/mL showed better cell protection. The three nanostructured drugs had no effect on pancreatic β-cell survival compared to the control group.

Subsequently, we used flow cytometry to detect the distribution of DNA content in INS-E cells of each drug treatment group to further clarify the intracellular cycle arrest before apoptosis. Compared to the control group, the DNA content in the G2M phase of the PA group was significantly increased, reaching 35.28%, while the DNA content in the G2M phase of the GLSO@NE, SeNP, and GLSO@SeNPs groups was reduced (Figure 3C,D). In particular, 0.04 μL/mL GLSO@SeNPs significantly reduced the proportion of cells in cell cycle arrest to 16.79%. In addition, the SubG1 content is the most direct indicator of apoptosis [36]. Compared to the control group, the peak content of SubG1 in the PA group was significantly increased, reaching nearly 30%, whereas treatment with 0.04 μL/mL GLSO@SeNPs significantly reduced the peak content of SubG1 (5.22%) compared to treatment with GLSO@NEs and SeNPs (Figure 3E). We further observed the cell morphology of each group using an electron microscope (Figure 3F). Cells in the control group grew normally in clumps and were adhered to the wall with irregular angular edges, whereas cells in the PA group showed a large wrinkled area and floated in the culture medium. Compared to the PA group, the survival rate of INS-1E cells in the three nanoprotective drug intervention groups was significantly improved, especially the PA + GLSO@SeNPs group.

### 3.4. GLSO@SeNPs Reverse PA–Induced Mitochondrial Dysfunction

Mitochondria play an important role in regulating apoptosis. Therefore, we investigated whether PA-induced apoptosis is related to mitochondrial dysfunction. Figure 4A shows that PA induced a significant increase in the proportion of intracellular monomer groups (green fluorescence groups), reaching 30.85%, while GLSO@SeNPs significantly reduced the proportion to 7.73%. In addition, Figure 4B shows that PA induced a significant reduction of nearly 2% in the proportion of red-green fluorescence groups in cells, while GLSO@SeNPs significantly increased the proportion to 12%. In contrast, pretreatment with GLSO@SeNPs effectively repaired PA-induced mitochondrial fracture. As shown in Figure 4C,D, the mitochondria in the control group showed elongated filaments evenly spread throughout the cytoplasm, while the fractured mitochondria induced by PA treatment showed spot-like fragments distributed in cells. In addition, GLSO@SeNPs intervention reversed the damage to some extent.

### 3.5. GLSO@SeNPs Reduce PA-Induced High Intracellular ROS Levels

Because ROS levels are indicators of oxidative stress in organisms, we used the fluorescein-labeled dye, DCFH-DA, to detect intracellular ROS. Compared to the control group, 0.3 mM PA treatment for 12 h induced a significant increase in intracellular ROS in INS-1E cells, but GLSO@SeNPs pretreatment significantly decreased the intracellular ROS production before exposure to PA (Figure 5A). Specifically, 0.04 μL/mL GLSO@SeNPs reduced the intracellular ROS content from 130% to 110% in the PA group. In addition, INS-1E cells treated with 0.04 μL/mL GLSO@SeNPs alone showed no significant difference in the intracellular ROS content compared to the control. After PA treatment, the green fluorescence level of the DCFH-DA label in INS-1E cells was significantly increased (Figure 5B). Pretreatment with 0.04 μL/mL GLSO@SeNPs significantly reduced the fluorescence level in PA-treated cells, but GLSO@SeNPs had no effect on the accumulation of free radicals in INS-1E cells.

### 3.6. GLSO@SeNPs Regulate Antioxidant Enzyme Activity

The redox system also plays a key role in the regulation of apoptosis. Because the PA-induced apoptosis in INS-1E cells was mainly due to increased oxidative stress, we detected the activity of redox-balancing enzymes in cells. Compared to the control group, the activities of SOD, GSH, GPX, TrxR, and CAT in INS-1E cells were significantly decreased by PA treatment (Figure 6A–E), but pretreatment with 0.04 μL/mL GLSO@SeNPs significantly protected the activities of these enzymes from the effects of PA and restored them to the level of the control group. Malondialdehyde (MDA), the final product of lipid oxidation, affects the mitochondrial respiratory chain complex and the activities of key enzymes in mitochondria in vitro. As shown in Figure 6F, PA treatment significantly increased the intracellular MDA content, and 0.04 μL/mL GLSO@SeNPs significantly inhibited MDA production, reducing it to the level of the control group. In addition, PA treatment decreased the TrxRI and TrxRII enzyme activities, and although GLSO@SeNPs tended to increase the activity of TrxR1 and TrxR2, there was no significant difference in the protective effect (Figure 6G,H).

### 3.7. Metabolism of GLSO@SeNPs in INS-1E Cells

To further understand how GLSO@SeNPs play a protective role after entering INS-1E cells, we tested the intracellular metabolism of GLSO@SeNPs using HPLC-ICP-MS. Selenium is stored in the tissue in the form of organic salt, which is conducive to participating in the biosynthesis of GSH-Px, while its inorganic salt is not stored. In the body, selenium mainly exists in the form of selenium and protein-binding complexes, and the protein containing selenocysteine residues is called selenoprotein. In nature, there are five valence states of selenium, including selenide (Se 2−), elemental selenium (Se), selenite (Se 2+), selenite (Se 4+), and selenate (Se 6+). As shown in Figure 7A,B, almost no selenium metabolites were detected in the cell protein extracts of the control group, while the content of SeCys2 in INS-1E cells was significantly increased to 0.03 and 0.02 μg/5 × 10^6^ cells after treatment with GLSO@SeNPs and GLSO@SeNPs + PA for 12 h, respectively. In addition, the content of SeCys2 in the GLSO@SeNPs + PA group was lower than that in the GLSO@SeNP group, which confirmed that PA treatment inhibited the GLSO@SeNP absorption efficiency of cells, resulting in cell damage. Moreover, after treatment with GLSO@SeNPs and GLSO@SeNPs + PA for 24 h, the content of SeCys2 in INS-1E cells increased to 0.075 and 0.086 μg/5 × 10^6^ cells, respectively. At this time, the content of SeCys2 in the GLSO@SeNPs + PA group was almost the same as that in the GLSO@SeNP group, which indicated that GLSO@SeNPs eliminated PA-induced cell arrest with the doubling of time. GLSO@SeNPs are transformed from a zero-valent state to SeCys2 through biological metabolism, thus effectively inhibiting PA-induced pancreatic cytotoxicity.

### 3.8. GLSO@SeNPs Regulate the ROS-Mediated Mitogen-Activated Protein Kinase (MAPK) Signaling Pathway

To further explore the potential mechanism of apoptosis induced by high ROS levels, we used western blot analysis to detect the expression of proteins related to the MAPK signaling pathway. Compared to the control group, treatment with 0.3 mM PA significantly increased the phosphorylation of p38 and JNK, while pretreatment with 0.04 μL/mL GLSO@SeNPs significantly reduced PA-induced phosphorylation of p38 and JNK (Figure 8A,B). In addition, GLSO@SeNPs pretreatment did not significantly affect the expression of p38 and JNK. As a key protein in the MAPK pathway, ERK plays a key regulatory role in cell growth and proliferation. Compared to the control group, the phosphorylation level of ERK induced by PA was significantly decreased (0.15 times), and pretreatment with 0.04 μL/mL GLSO@SeNPs significantly increased the ERK phosphorylation level and promoted the growth and proliferation of cells. In addition, AKT indirectly promotes cell growth and proliferation by inhibiting the expression of proapoptotic proteins. Therefore, we investigated whether GLSO@SeNPs regulate PA-induced AKT inhibition. Compared to the control group, the expression of phosphorylated AKT in INS-1E cells was significantly decreased after treatment with 0.3 mM PA, whereas pretreatment with GLSO@SeNPs significantly increased the expression of phosphorylated AKT, which demonstrated the importance of AKT in regulating PA-induced growth inhibition.

### 3.9. GLSO@SeNPs Suppress PA-Induced Caspase Activation

To explore the potential mechanism of the protective effect of GLSO@SeNPs on PA-induced apoptosis of INS-1E cells, we used western blot analysis to detect the expression levels of Bcl-2, caspase family proteins, and PARP cleavage in PA-induced INS-1E cells. PARP cleavage is also one of the induction factors of apoptosis, which is located downstream of caspase 3 in the apoptotic pathway [37]. As shown in Figure 9A,B, treatment with 0.3 mM PA significantly activated caspases 3, 8, and 10, resulting in decreased protein expression levels at corresponding sites. The activities of caspases 3, 8, and 10 were significantly inhibited after treatment with 0.04 μL/mL GLSO@SeNPs for 6 h. In addition, PA induced the expression of proapoptotic proteins, including Bax, Bim, and Bad, but inhibited the expression of the antiapoptotic protein, Bcl-2, in INS-1E cells, which suggested that PA initiated the mitochondrial apoptosis signaling pathway. Treatment with 0.04 μL/mL GLSO@SeNPs for 6 h inhibited the expression of proapoptotic proteins.

## 4. Discussion

Overall, the present study was based on the synthesis of nanomaterials with high antioxidant activity. After the morphology characterization and stability test of the nanomaterials, they were applied in diabetes research. According to previously reported studies [38,39,40], we selected the common model of PA-inducing apoptosis of INS-1E pancreatic cells, and we pretreated the cells with a certain dose of nanomaterials. By detecting cell viability, cycle distribution, mitochondrial membrane potential, ROS, oxidative kinase content, and apoptotic protein signaling pathways, we demonstrated that SeNPs, GLSO@NEs, and GLSO@SeNPs prevent and improve INS-1E cytotoxicity caused by PA. In particular, GLSO@SeNPs show the highly synergistic antioxidant activity of SeNPs and GLSO@NEs, thus protecting cells from apoptosis to the greatest extent. Finally, western blot analysis demonstrated that GLSO@SeNPs protect INS-1E cells from PA lipotoxicity by upregulating or inhibiting the expression of some key proteins in the MAPK signaling pathway and inhibiting the oxidative stress-mediated apoptosis pathway.

Using GLSO with antioxidant activity and nanoselenium as the active components, we designed and synthesized a GLSO@SeNP hydrogel composite system through emulsion preparation, high-pressure homogenization, and other technologies, differing from previously reported studies on NPs [41,42,43]. In the present study, NPs of the same system were simultaneously compared. For the same cytotoxicity model, it was concluded that the composite material had more significant antioxidant activity than either single material alone. Raman spectroscopy demonstrated that GLSO@SeNPs had activity characteristic peaks of both SeNPs and GLSO@NEs, indicating that GLSO@SeNPs combined the antioxidant active sites of SeNPs and GLSO@NEs (Figure 1D). The long-term stability of GLSO@SeNPs in various physiological solutions, especially human blood, benefited the good antiapoptotic activity (Figure 2). It is well known that evaluating the stability of NPs is of great significance to determine whether they have potential biological protective effects. In the present study, the characterization tests (Figure 1A–C) confirmed that the synthesized NPs were within reasonable and reliable size and potential ranges.

Diabetes mellitus can be divided into type 1 diabetes mellitus and type 2 diabetes mellitus. The etiology of type 1 diabetes mellitus is the obstruction of insulin secretion of pancreatic cells, which leads to the unregulated high blood sugar level of the body, whereas type 2 diabetes mellitus is manifested by congenital insulin resistance or insufficient insulin secretion. However, the long-term increase of free fatty acids represented by PA leads to the apoptosis of pancreatic β-cells. As a result, the function of insulin secretion is reduced [44,45,46]. Therefore, pancreatic β-cell retention may be an effective strategy to prevent diabetes. Previous studies have shown that rosiglitazone and other natural active ingredients derived from animals and plants can maintain the vitality of pancreatic INS-1E cells to effectively prevent diabetes [47,48,49]. In addition, the use of PA, as the simulation environment for hyperlipemia, induced lipid toxicity to a greater extent, and the concentration of 0.3 mM PA was similar to the hyperlipemia state produced by the occurrence of diabetes in the body. The culture of INS-1E cells with PA for 24 h more realistically simulated the long-term effects of lipid toxicity. Of note, apoptosis, also known as programmed cell death, refers to the orderly death of cells under the control of related genes to maintain the stability of the internal environment of the body. The process of apoptosis includes the activation, expression, and regulation of proteins, indicating that it is an active death process. In the present study, PA induced the long-term arrest of INS-1E cells in the G2/M phase of mitosis, which interrupted the next nucleic acid DNA synthesis cycle, resulting in cell growth arrest and eventual apoptosis. In the present study, GLSO-functionalized selenium NPs were used for the first time to induce apoptosis of INS-1E cells to generate an apoptotic model of pancreatics, and it was concluded that these NPs have antiapoptotic activity. The cytotoxicity and cycle analyses (Figure 3A–F) demonstrated that GLSO@SeNPs inhibited apoptosis in a concentration-dependent manner and that GLSO@SeNPs significantly reduced G2/M arrest, confirming that GLSO@SeNPs maintain cell viability by promoting intracellular DNA synthesis. In addition, the cell growth activity in the GLSO@SeNP treatment group alone increased compared to the control group, further indicating that GLSO@SeNPs have an antiapoptotic effect in INS-1E cells.

Mitochondria are organelles that continuously supply energy to cells and are closely related to the occurrence of cell apoptosis, and mitochondrial dysfunction and morphological damage are common in metabolic syndrome diseases, such as diabetes [50,51]. In the present study, the destructive power of PA on mitochondria and the protective effect of GLSO@SeNPs on mitochondria were evaluated by detecting mitochondrial membrane potential and mitochondrial fragmentation. Experimental evidence showed that GLSO@SeNPs reversed the mitochondrial membrane potential decline and morphological damage induced by PA, thereby maintaining the normal operation of mitochondria in INS-1E cells (Figure 4A–C). Previous studies have shown that oxidative stress is the key cause of cell apoptosis, and stimulation of external fatty acids causes accumulation of ROS in the body, thus attacking macromolecular substances and cell walls, which cause cell body damage and lesions, accelerating the process of apoptosis [52]. Therefore, inhibiting ROS production is of great significance for alleviating cell apoptosis. In the present study, total intracellular ROS levels were detected using DCFH-DA. The experimental results showed that ROS levels were significantly increased in the PA group, with green fluorescence in large areas of cells (Figure 5A,B). The strong antioxidant activity of GLSO@SeNPs significantly removed intracellular ROS, thus alleviating high intracellular oxidative stress.

SOD is the primary enzyme that scavenges free radicals in organisms, and it converts superoxide anion free radicals into H_2_O_2_ and O_2_. CAT and GSH-Px decompose H_2_O_2_ into H_2_O and O_2_, while GPX reduces H_2_O_2_ into non-toxic hydroxyl compounds. TrxR has a similar activity to glutathione reductase (GR). Catalytic reduction of glutathione disulfide (GSSG) to GSH is one of the key enzymes in the glutathione redox cycle [53]. Under the action of PA, the activity of antioxidant enzymes in pancreatic β-cells is greatly affected, and MDA, the final product of lipid peroxidation, is produced in large quantities, which leads to the weakening of the free-radical-scavenging ability of β-cells and then apoptosis [54]. In the present study, the contents of SOD, GSH, GPX, CAT, and TrxR in cells were detected, which demonstrated that PA significantly inhibited the activity of antioxidant enzymes in cells, induced the production of a large amount of MDA, and increased the degree of lipid peroxidation in the body, thus leading to cell damage (Figure 6A–H). However, the use of GLSO@SeNPs enhanced the activity of these antioxidant enzymes and eliminated MDA, confirming that GLSO@SeNPs have the ability to regulate enzyme activity to prevent PA-induced pancreatic β-cells lipotoxicity and apoptosis. We believe that antioxidant enzymes are important participants in the cellular redox system, and it is particularly important to maintain good, efficient enzyme activity to maintain the homeostasis of the cell environment. Therefore, GLSO@SeNPs in this study significantly improved the activity of SOD, TrxR, CAT, GSH, and GPx. These results showed that it has a protective effect on oxidative-stress-mediated apoptosis.

Many drugs have been shown to exert their biological activity through blood transport to target organs and metabolism by the gut microbiome [55]. To further verify the mechanism of GLSO@SeNPs protecting INS-1E cells from lipid toxicity, we evaluated the intracellular metabolites of GLSO@SeNPs. The present findings indicated that GLSO@SeNPs were transformed from the initial zero-valent state into the SeCys2 form, resulting in a protective effect. The present study also confirmed that selenium-containing drugs acted in cells in the form of selenoproteins (Figure 7A,B). In addition, we believe that the participation of GLSO plays a role in modifying SeNPs and promoting the efficient absorption and release of selenium in INS-1E cells. Such functionalized selenium nanoparticles can effectively inhibit the islet lipid toxicity induced by exposure to free fatty acids.

Previous studies have shown that ROS-induced oxidative stress stimulates and activates the MAPK signaling pathway in tissue and cells, and there is a close correlation between the two pathways [56]. The MAPK signaling pathway regulates cell growth, differentiation, environmental stress adaptation, inflammatory response, and other important cell physiological processes. Exposure to ultraviolet (UV) radiation or saturated free fatty acids activates ERK, JNK, and p38, leading to oxidative stress and apoptosis in tissues [57]. Mitochondria are the main source of ROS production, and mitochondrial dysfunction is an important event inducing apoptosis. The integrity of the mitochondrial outer membrane is regulated by Bcl-2 family members. The isomerization of dimers related to the mitochondrial outer membrane, such as Bax/Bak, permeabilizes the mitochondrial membrane, resulting in the release of cytochrome C from the intermembrane into the cytoplasm. Cytochrome C then acts synergistically with Smac and induces caspase activation [58]. In the present study, western blot analysis demonstrated that GLSO@SeNPs reversed apoptosis by inhibiting the ROS-mediated apoptosis pathway via inhibiting ROS production by downregulating p-ERK and p-AKT, as well as upregulating p-p38 and p-JNK (Figure 8A–C). In addition, GLSO@SeNPs prevented apoptosis by inhibiting the activation of caspases 3, 8, and 10, as well as the expression of the Bax, Bad, Bim, and C-PARP [59] proapoptotic proteins. Therefore, the present study suggested that GLSO@SeNPs significantly inhibit PA-induced cytotoxicity of INS-1E cells, thereby improving their viability (Figure 9A–C).

## 5. Conclusions

In summary, the present results suggested that a certain dose of GLSO@SeNPs cells protects INS-1E cells from PA-induced oxidative damage by inhibiting oxidative stress-mediated apoptosis pathways and reversing PA-induced antioxidant enzyme inactivation, thereby improving cell survival. Although the present results supported the beneficial potential of GLSO-functionalized selenium NPs in the prevention and treatment of lipotoxic-induced diabetes, several limitations of the present study need to be addressed. In the present study, only PA-induced cytolipotoxicity in vitro was used to demonstrate that GLSO@SeNPs inhibit INS-1E cell apoptosis. In the future, animal experiments should be conducted to verify the therapeutic effect of GLSO@SeNPs on diabetes in vivoin vivo, and the potential mechanism of the protective effect of GLSO@SeNPs should be comprehensively explored through pharmacokinetics and metabolomics to further expand their biological application.

## Figures and Tables

**Figure 1 antioxidants-12-00840-f001:**
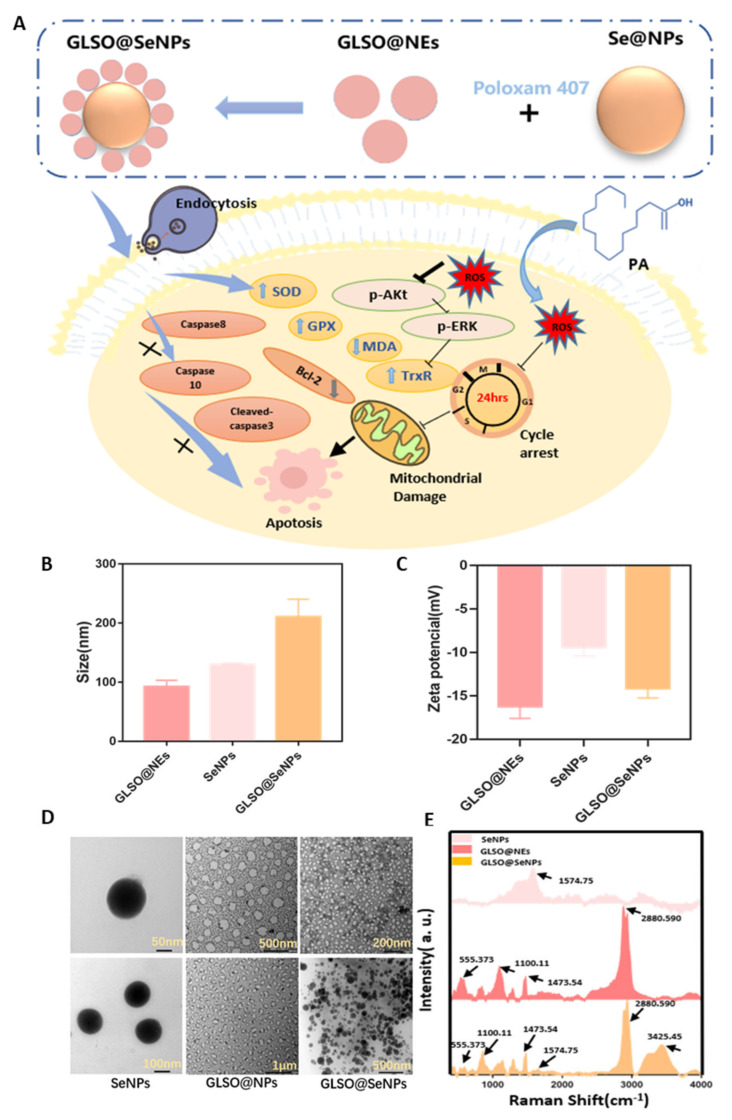
Characterization of SeNPs, GLSO@NEs, and GLSO@SeNPs. (**A**) Schematic showing the protective effect of GLSO@SeNPs on PA—induced apoptosis of INS-1E cells. Particle size (**B**), potential values (**C**), TEM images (**D**), and Raman spectra (**E**) of SeNPs, GLSO@NEs, and GLSO@SeNPs.

**Figure 2 antioxidants-12-00840-f002:**
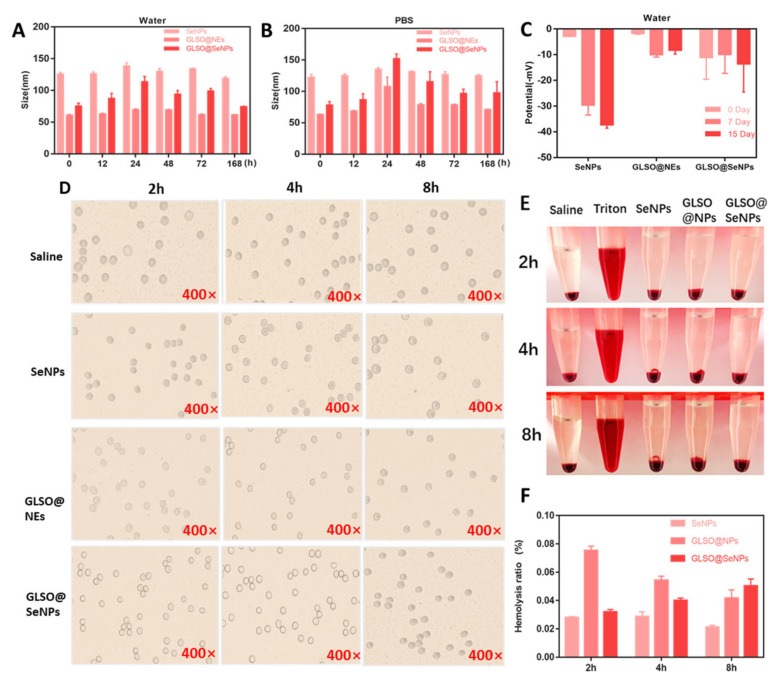
Stability assessment of SeNPs, GLSO@NEs, and GLSO@SeNPs. Changes in particle size of SeNPs, GLSO@NEs, and GLSO@SeNPs in water (**A**) and PBS (**B**) during 168 h. (**C**) Changes in Zeta potential of SeNPs, GLSO@NEs, and GLSO@SeNPs in water over a 15-day period. After incubating SeNPs, GLSO@NEs, and GLSO@SeNPs with human blood for 2, 4, and 8 h, the morphology of red blood cells was observed under a microscope (400×) (**D**) and by the naked eye (**E**). (**F**) Hemolysis rates of SeNPs, GLSO@NEs, and GLSO@SeNPs at 2, 4, and 8 h.

**Figure 3 antioxidants-12-00840-f003:**
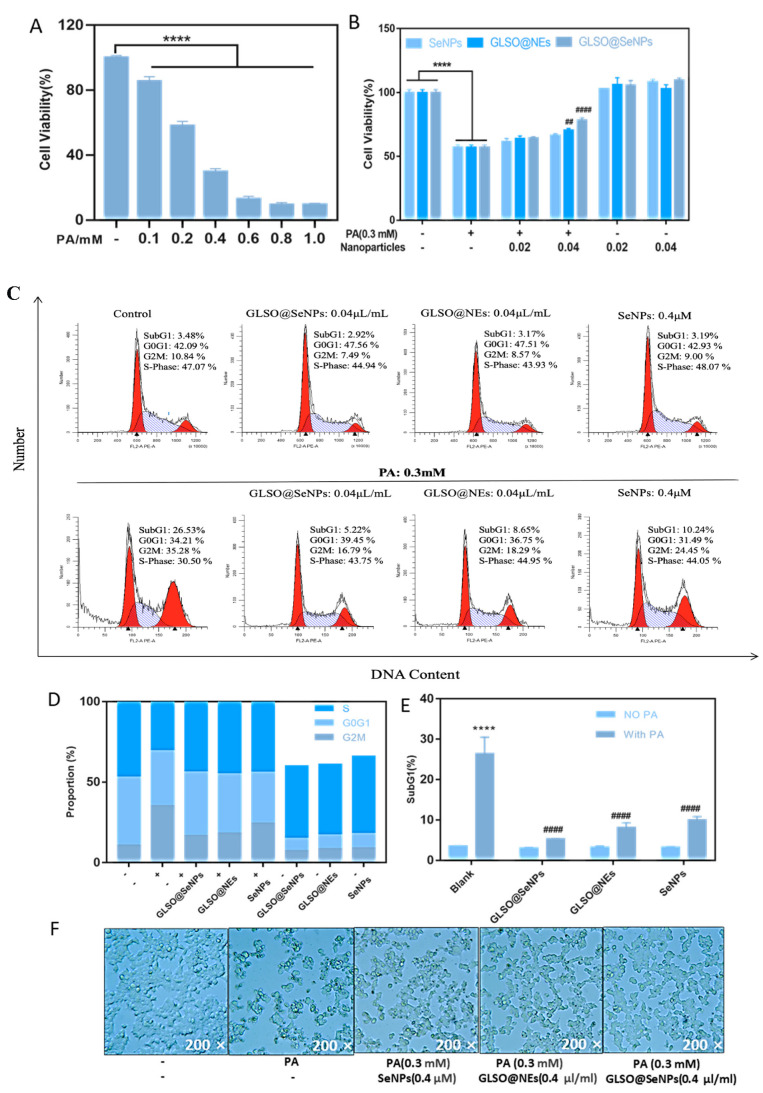
Protective effects of NPs. (**A**) Cell viability after treatment with different concentrations of PA for 24 h. (**B**) The effects of different concentrations of SeNPs, GLSO@NEs, and GLSO@SeNPs on the survival rate of INS-1E cells after 6 h of early intervention followed by PA treatment. (**C**) Modfit software (5.0) was used to analyze the distribution of INS-1E cells in each cycle. (**D**) Quantitative analysis of cell cycle distribution in each group. (**E**) Quantitative analysis of the SubG1 peak in INS-1E cells. (## *p* ˂ 0.01 means moderate significance, **** (####) *p* ˂ 0.0001 means the highest significance.) (**F**) Morphology of INS-1E cells treated with different NPs as observed under a microscope (200×).

**Figure 4 antioxidants-12-00840-f004:**
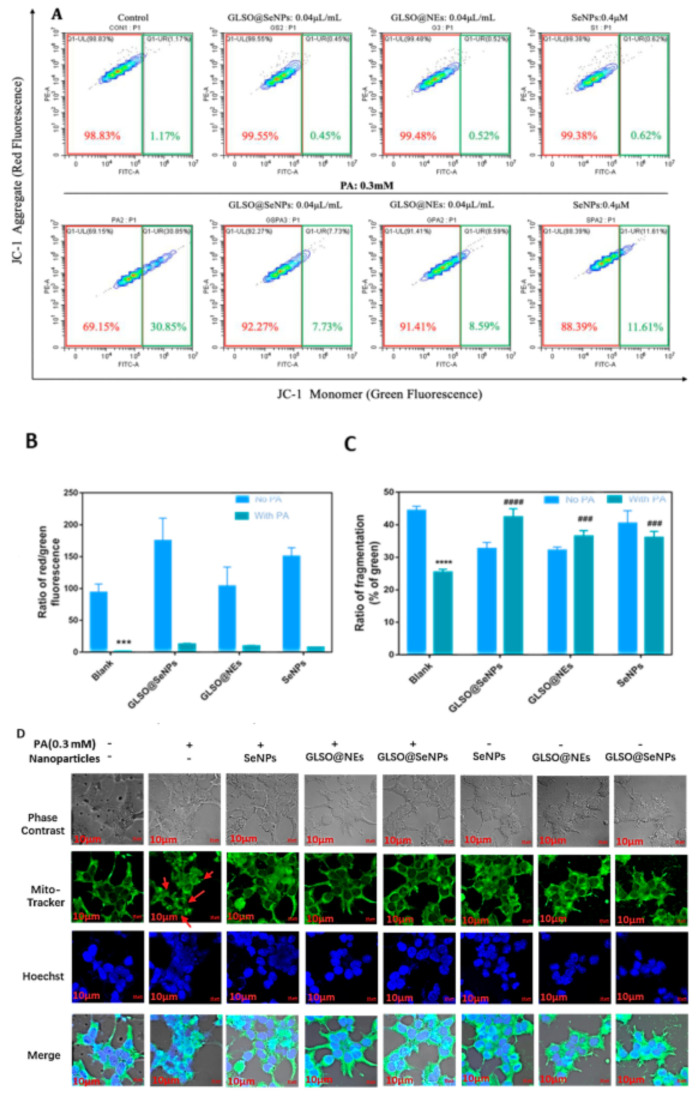
GLSO@SeNPs inhibit the PA-induced mitochondrial membrane potential decrease and fragmentation in INS-1E cells. (**A**) Distribution ratio of red and green fluorescence in INS-1E cells. (Red fluorescence refers to the JC-1 probe existing in the mitochondria of normal cells in the red polymer state, and green fluorescence refers to the JC-1 probe diffused from the mitochondria of apoptotic cells in the form of green monomer) (**B**) Quantitative analysis of the ratio of red/green fluorescence in INS-1E cells. (**C**) Quantitative analysis of mitochondrial fragmentation by fluorescence image. (*** (###) *p* ˂ 0.05 means high significance, **** (####) *p* ˂ 0.05 means the highest significance.) (**D**) Fluorescence images of mitochondrial disruption in cells of each drug treatment group.

**Figure 5 antioxidants-12-00840-f005:**
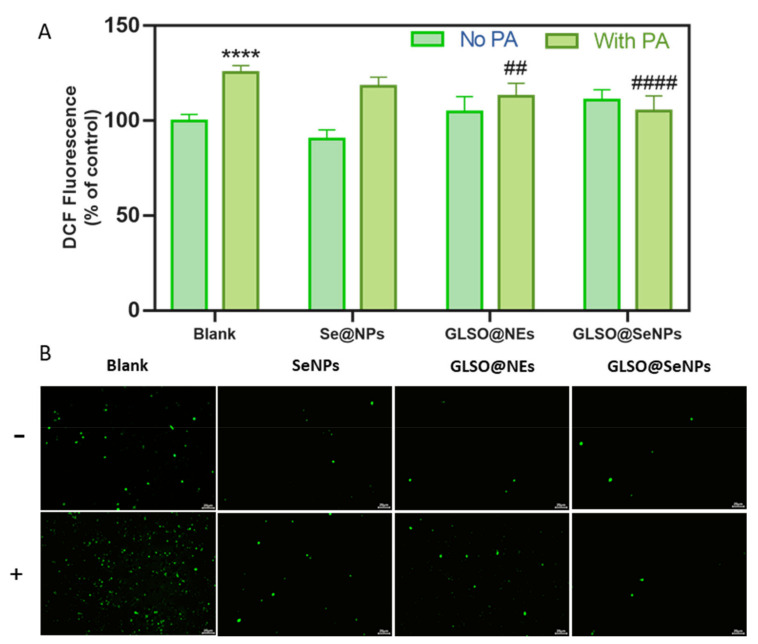
GLSO@SeNPs prevent PA-induced ROS generation in INS-1E cells. (**A**) ROS production changes in INS-1E cells after incubation with 0.04 μL/mL GLSO@SeNPs, 0.04 μL/mL GLSO@NEs, and 0.4 mM SeNPs for 6 h followed by PA stimulation for 12 h (cells were stained with DCFH-DA for 20–30 min; n = 6). (## *p* < 0.01 means moderate significance, **** (####) *p* < 0.0001 means the highest significance.) (**B**) Representative DCFH-DA fluorescence images of INS-1E cells exposed to PA after treatment with the different nanosystems.

**Figure 6 antioxidants-12-00840-f006:**
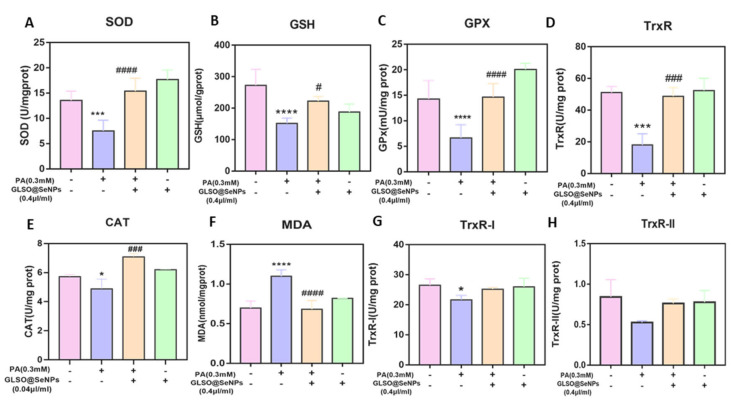
Determination of oxidative stress-related enzymes in cells. SOD content (**A**), GSH content (**B**), GPX activity (**C**), TrxR activity (**D**), CAT activity (**E**), MDA content (**F**), and TrxR-I activity in mitochondria (**G**), as well as TrxR-II activity in intracellular cytoplasmic proteins (**H**) in INS-1E cells of each drug treatment group. (* (#) *p* < 0.05 means the lowest significance, *** (###) *p* < 0.001 means high significance, **** (####) *p* < 0.0001 means the highest significance.)

**Figure 7 antioxidants-12-00840-f007:**
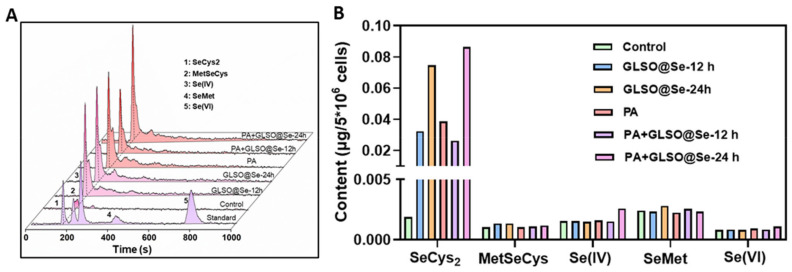
GLSO@SeNP metabolism in INS-1E cells. (**A**) Selenium metabolites, including SeCys2, MetSeCys, Se(IV), SeMet, and Se(VI), in INS-1E cells after treatment for 12 and 24 h. (**B**) Quantitative analysis of selenium metabolites.

**Figure 8 antioxidants-12-00840-f008:**
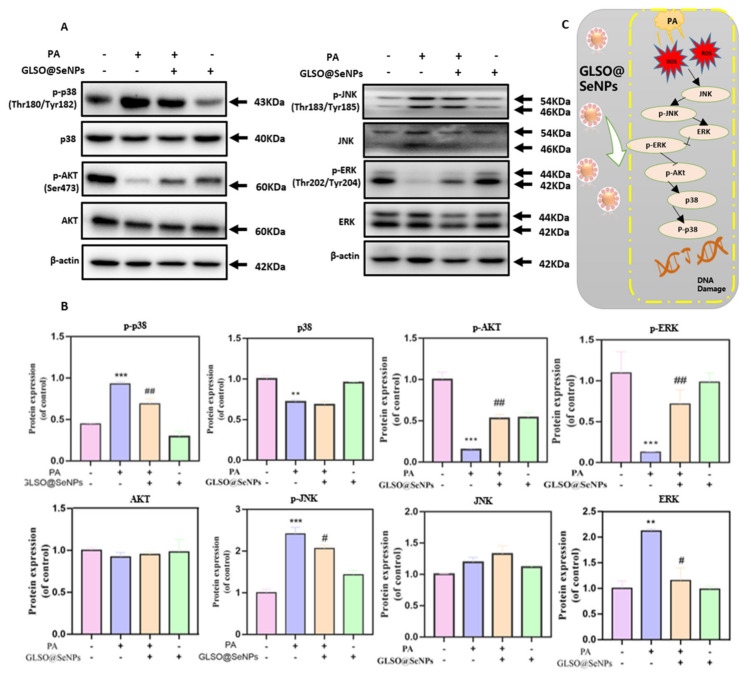
GLSO@SeNPs inhibit PA-induced apoptosis by regulating the MAPK signaling pathway in INS-1E cells. (**A**) Protein expression levels of p-p38, p38, p-AKT AKT, p-JNK, JNK, p-ERK, and ERK. (**B**) Quantitative analysis of protein bands. (Firstly, the grayscale of the target strip was divided by the grayscale of the corresponding β-actin reference strip, and then the mean value of the control group was calculated. Finally, each group was divided by the mean value of the control group. The resulting values are used to make bar charts.) (# *p* < 0.05 means the lowest significance, ** (##) *p* < 0.01 means moderate significance, *** *p* < 0.001 means high significance). (**C**) Schematic diagram of the MAPK signaling protein pathway.

**Figure 9 antioxidants-12-00840-f009:**
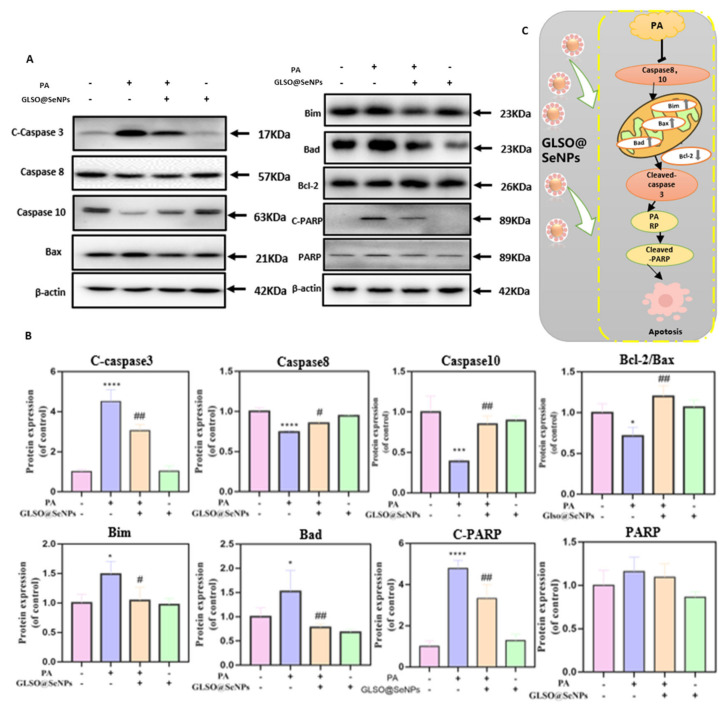
GLSO@SeNPs attenuate PA-induced apoptosis by inhibiting caspase activation in INS-1E cells. (**A**) Protein expression levels of c-caspase 3, caspase 8, caspase 10, Bax, Bim, Bad, C-PARP, and PARP. (**B**) Quantitative analysis of protein bands. (Firstly, the grayscale of the target strip was divided by the grayscale of the corresponding β-actin reference strip, and then the mean value of the control group was calculated. Finally, each group was divided by the mean value of the control group. The resulting values are used to make bar charts.) (* (#) *p* < 0.05 means the lowest significance, ## *p* < 0.01 means moderate significance, *** *p* < 0.001 means high significance, **** *p* < 0.0001 means the highest significance.) (**C**) Schematic diagram of the apoptosis signaling protein pathway.

## Data Availability

The data are contained within this article.

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
