# Peer review of "Spore Oil-Functionalized Selenium Nanoparticles Protect Pancreatic Beta Cells from Palmitic Acid-Induced Apoptosis via Inhibition of Oxidative Stress-Mediated Apoptotic Pathways"

_antioxidants, 2023, doi:10.3390/antiox12040840_

Round 1

Reviewer 1 Report

This study investigates the link between oxidative stress damage of pancreatic β cells and type 2 diabetes mellitus, and shows that Ganoderma lucidum spore oil (GLSO) and GLSO-functionalized selenium nanoparticles (GLSO@SeNPs) effectively inhibit palmitic acid (PA)-induced apoptosis of pancreatic cells by scavenging ROS and improving antioxidant enzyme activity. GLSO@SeNPs protect INS-1E β cells from PA-induced toxicity by inhibiting intracellular oxidative stress levels and could be used as a new nutritional compound drug to antagonize PA-induced pancreatic damage. However, in its present stage, the manuscript presents deficiencies that should be revised to reach the possibility of publication.
1. In Figure 4(A), What do the red and green fluorescence mean? A more detailed figure legend explanation is needed.
2. You should indicate the quantitative analysis of fluorescence images of the mitochondrial disruption 

3. In Figures 8B and 9B, A detailed explanation of statistical analysis is needed. 

Reviewer 2 Report

This manuscript presented the result of examining the effect of Ganoderma lucidum spore oil (GLSO) and a synthetic selenium nanoparticle (GLSP@SeNPs) on the oxidative stress inhibition of pancreatic β-cells by palmitic acid. Research to find substances and compounds involved in the regulation of pancreatic beta cells and to verify their effects is very important. However, it is necessary to determine if the experiment was performed correctly and really and to publish the paper based on reliable experimental data and correct understanding. This manuscript is inadequate in that respect, and it is considered that it has not reached the stage where the research content can be peer-reviewed. The following are some of the questions.

comments

1.

Line 62-63: At present, studies have demonstrated that Ganoderma lucidum extract has a positive effect on serum metabolites of rats with type 2 diabetes [15].

It is necessary to specifically state what a "positive effect" is.

2.

Line 141-143: It is necessary to show specific measurement methods for TEM and Raman measurements. In addition to the measurement conditions of the sample, it is necessary to show in detail the conditions of the equipment at the time of measurement and the measurement settings such as laser intensity.

3.

Line 183-193: Regarding the analysis of cell cycle distribution, it is necessary to specifically show the experimental method and evaluation criteria for judging G1, G2, and S-Phase.

4.

Line 195-199: Reagents and methods used for mitochondrial membrane potential should be indicated.

5.

Line 240-241: It is necessary to indicate the specific method of HPLC (column, solvent, speed, etc.).

6.

Line 246-250: Please indicate the difference between # and * in the text. It is also necessary to indicate the meaning of ### and ***.

7.

Line 266: It is necessary to cite the reason for judging as reasonable or the paper supporting the interpretation.

8.

According to the graph in Fig. 1A, the particle size of each sample varies. It is necessary to change the notation to the average value and its variation (standard deviation, etc.).

9.

The cartoon at the top of Figure 1 should also be numbered and accompanied by a figure legend. Corresponding parts must be specified in the text. Some letters are underlined in red. The authors should explain the authors' intentions.

10.

This reviewer can't see the scale in Figure 1c. Figure resolution needs to be improved.

11.

Line 276-277: Need to cite a paper that confirms that 1574.75 cm-1 is the Lentinan peak.

12.

Line 283-285: It is stated that GLSO@SeNPs has characteristic peaks of both SeNPs and GLSO@NPs. There is no notation of the peak in the graph. A new peak appears at 3425 cm-1. It is necessary to specify information such as the notation of characteristic peaks and the molecular structure from which the peaks are derived.

13.

Line 296-297: The paper should be cited for "The particle size and potential are two main indexes used to evaluate the stability of a NP".

14.

Line 299-302: It is necessary to indicate the degree of variability, such as by notation of mean ± SD (min~max).

15.

Figure 2A-C: The authors need to state the reason for the timeline. GLSO@SeNPs appear to change size after 24h. Reasons and considerations for the change should be given, as to why such size differences occur.

16.

The morphology cannot be determined from Figure 2D. Figure resolution needs to be improved.

There is no scale bar, and the GLSO@SeNPs-supplemented cells also appear to be boxed in red. The authors need to explain what this red line is.

17.

Line 331-332: Please explain why you chose 0.3mM.

18.

Figure 3A: The authors need to explain why they did the statistical analysis compared to the 0.1mM sample. Consider if there was a significant difference at 1mM only, in which case the concentration used should be 1mM.

19.

Figure 3B, D: The authors need to explain how the significance tests in the figures were done. Marks need to be explained.

20.

Figure 3: The order explained in the text and the order of the figures are different. It needs to be revisited for better understanding.

21.

Figure 3F: Need to add scale bar.

22.

Lines 341-342: The text and Figure 3B seem to give inconsistent results. Figure 3B shows that the GLSO@NEs group has a higher survival rate improvement effect when PA was added than the GLSO@SeNP group.

23.

Line 350-351: The paper should be cited for In addition, the SubG1 content is the most direct indicator of apoptosis.

24.

Line370-371: This reviewer doesn't understand what "Green Fluorescence group" and "Ref Fluorescence" indicate. The authors need to explain what the Red/Green ratio indicates.

25.

Figure 4C: The authors should clearly state what the -/+ in the figure means. This reviewer can't read the scale of the figure.

26.

Figure 4: Changes in mitochondrial fragmentation are not clear. It is necessary to give an arrow to indicate which part in the diagram is pointing.

27.

The results in Figures 5A and 5B are inconsistent. The results in Figure 5B show that the addition of SeNPs, GLSO@Nes, and GLPS@SeNps to blanks without PA decreased the number of cells exhibiting DCFH-DA fluorescence. However, the text indicates that GLPS@SeNPs alone did not change (lines 395-396). Results need to be properly understood and stated accurately.

28.

Figure 6: The order of the figures differs from the order described in the text. It needs to be revisited for better understanding.

29.

Figure 6: The authors need to show the definition of ACTIVITY, and specify what they mean by "mg prot".

30.

Figure 6E: The addition of GLSP@SeNPs appears to increase CAT (U/mg prot). The result and text do not match (Line 418-419). Results need to be properly understood and stated accurately.

31.

Figure 6B: Does GLSP@SeNPs addition significantly (####) increase MDA levels after PA addition? Results need to be properly understood and stated accurately.

32.

Line425-426: Why check TrxR1 and TrxR2? What is meant by "a certain extent" needs to be clarified. The results of statistical analysis and their understanding should be presented.

33.

Line 450-451: Insufficient explanation and consideration of the significance of the change of 0.01ug.

34.

Figure 8c: This reviewer is not sure what the green arrow is for. Also, the results of GLSO@SeNPs directly inhibiting PA have not been shown.

35.

Line495496: A paper should be cited for ``PARP cleavage is also one of the induction factors of apoptosis, which is located downstream of caspase 3 in the apoptotic pathway.''

36.

Figure 9A: The authors need to explain the intent of the red line.

37.

Similar to Figure 8C, Figure 9C needs to be corrected.

38.

Figure 8A, B: The expression level of caspase8,10 is decreased by PA addition, and the results are inconsistent with the text (Line 500-501). Results need to be properly understood and stated accurately.

39.

The authors need to explain the difference between C-PARP and PARP.

40.

Does GLSO@SeNPs improve the insulin secretory function of pancreatic β cells during PA-induced stress?

41.

There is no figure legend for the attached original images, and the explanation is insufficient.

Reviewer 3 Report

Review of the manuscript entitled: Spore oil-functionalized selenium nanoparticles protect pancreatic beta cells from palmitic acid-induced apoptosis via inhibition of oxidative stress-mediated apoptotic pathways. Most importantly, the manuscript is not formatted according to journal requirements. This disrespectful to the Journal!

The manuscript is interesting but some corrections should be made. In abstract clear aim of the manuscript should be added e.g. "The aim of the present study was to ...".

In introduction you can't describe your results, please remember it's not an abstract and should end with aim of manuscript. Therefore, lines 95 to 103 should be deleted or transfer to conclusion (or discussion).

Despite this, the introduction is generally good, but please note that Latin names should be in italics e.g. Ganoderma lucidum.

In the material and methods section, the catalog numbers of all key reagents should be listed. Please complete the antibody and commercial kit catalog numbers this is crucial if anyone wants to verify the results.

The results are described in detail but, in the description of the results, we do not provide references, and we do not discuss the results. The results are discussed in the discussion.

Therefore, sentences such as lines 286-289; 315-316; 340-352; 355-356 and so on should be removed from the description of the results and transferred to the discussion. Check carefully entire manuscript, all abbreviations should be explained.

Line 553 “rosiglidone” shouldn't it be rosiglitazone? Please check.

The discussion could be deeper but it is satisfactory. Concussion is correct.

Round 2

Reviewer 2 Report

The authors have successfully processed the manuscript in line with the reviewer's comments. Each answer is good and has been corrected with text and figures accordingly. Overall, this manuscript is suitable for readers interested in antioxidants.